# NECST: Neural Joint Source-Channel Coding

## Abstract

For reliable transmission across a noisy communication channel, classical results from information theory show that it is asymptotically optimal to separate out the source and channel coding processes. However, this decomposition can fall short in the finite bit-length regime, as it requires non-trivial tuning of hand-crafted codes and assumes infinite computational power for decoding. In this work, we propose Neural Error Correcting and Source Trimming (NECST) codes to jointly learn the encoding and decoding processes in an end-to-end fashion. By adding noise into the latent codes to simulate the channel during training, we learn to both compress and error-correct given a fixed bit-length and computational budget. We obtain codes that are not only competitive against several capacity-approaching channel codes, but also learn useful robust representations of the data for downstream tasks such as classification. Finally, we learn an extremely fast neural decoder, yielding almost an order of magnitude in speedup compared to standard decoding methods based on iterative belief propagation.

## 1 Introduction

We consider the problem of encoding images as bit-strings so that they can be reliably transmitted across a noisy communication channel. Classical results from (Shannon (1948)) show that for a memoryless communication channel, as the image size goes to infinity, it is optimal to separately: 1) compress the images as much as possible to remove redundant information (source coding) and 2) use an error-correcting code to re-introduce redundancy, which allows for reconstruction in the presence of noise (channel coding). This *separation theorem* has been studied in a wide variety of contexts and has also enjoyed great success in the development of new coding algorithms.

However, this elegant decomposition suffers from two critical limitations in the finite bit-length regime. First, without an infinite number of bits for transmission, the overall distortion (reconstruction quality) is a function of both source and channel coding errors. Thus optimizing for both: (1) the number of bits to allocate to each process and (2) the code designs themselves is an extremely difficult task. Second, maximum-likelihood decoding is in general NP-hard (Berlekamp et al. (1978)). Thus obtaining the accuracy that is possible *in theory* is contingent on the availability of infinite computational power for decoding, which is impractical for real-world systems. Though several lines of work have explored various relaxations of this problem to achieve tractability (Koetter & Vontobel (2003), Feldman et al. (2005), Vontobel & Koetter (2007)), many decoding systems rely on heuristics and early stopping of iterative approaches for scalability, which can be highly suboptimal.

To address these challenges we propose Neural Error Correcting and Source Trimming (NECST) codes, a deep learning framework for jointly learning to compress and error-correct an input image given a fixed bit-length budget. Three key steps are required. First, we use neural networks to encode each image into a suitable bit-string representation, sidestepping the need to rely on hand-designed coding schemes that require additional tuning for good performance. Second, we simulate a discrete channel *within* the model and inject noise directly into the latent codes to enforce robustness. Third, we amortize the decoding process such that after training, we can scale our model to massively large datasets. These components pose significant optimization challenges due to the inherent non-differentiability of discrete latent random variables. We overcome this issue by leveraging recent advances in unbiased low-variance gradient estimation for variational learning of discrete latent variable models, and train the model using a variational lower bound on the mutual information between the images and their binary representations to obtain good, *robust* codes. At its core, NECST

can also be seen as an implicit deep generative model of the data derived from the perspective of the joint-source channel coding problem.

In experiments, we test NECST on several grayscale and RGB image datasets, obtaining improvements over industry-standard compression (e.g, JPEG) and error-correcting codes (e.g., low density parity check codes). We also learn an extremely fast neural decoder, yielding almost an order of magnitude (two orders for magnitude for GPU) in speedup compared to standard decoding methods based on iterative belief propagation. Additionally, we show that in the process we learn discrete representations of the input images that are useful for downstream tasks such as classification.

## 2 SOURCE AND CHANNEL CODING

### 2.1 PRELIMINARIES

We consider the problem of reliably communicating data across a noisy channel using a system that can detect and correct errors. Let $\mathcal{X}$ be a space of possible inputs (e.g., images), and $p_{\text{data}}(x)$ be a *source* distribution defined over $x \in \mathcal{X}$. The goal of the communication system is to encode samples $x \sim p_{\text{data}}(x)$ as messages in a codeword space $\widehat{\mathcal{Y}} = \{0, 1\}^m$. The messages are transmitted over a noisy communication channel, where they are corrupted to become *noisy* codewords in $\mathcal{Y}$. Finally, a decoder produces a reconstruction $\hat{x} \in \mathcal{X}$ of the original input $x$ from a received *noisy* code $y$.

The goal is to minimize the overall *distortion* (reconstruction error) $\|x - \hat{x}\|$ in $\ell_1$ or $\ell_2$ norm while keeping the message length $m$ as small as possible (*rate*). In the absence of channel noise, low reconstruction errors can be achieved using a compression scheme to encode inputs $x \sim p_{\text{data}}(x)$ as succinctly as possible (e.g., JPEG). In the presence of channel noise, longer messages are typically needed to redundantly encode the information and recover from errors (e.g., parity check bits).

### 2.2 THE JOINT SOURCE-CHANNEL CODING PROBLEM

Based on fundamental results due to Shannon (Shannon, 1948), existing systems adhere to the following pipeline. A *source encoder* compresses the source image into a bit-string with the minimum number of bits possible. A *channel encoder* re-introduces redundancies, part of which were lost during source coding to prepare the codeword $\hat{y}$ for transmission. The *decoder* then leverages the channel coding scheme to infer the original signal $\hat{y}$, producing an approximate reconstruction $\hat{x}$.

In the separation theorem, Shannon proved that the above approach is optimal in the limit of infinitely long messages. That is, we can minimize distortion by optimizing the source and channel coding processes independently. However, given a finite bit-length budget, the relationship between rate and distortion becomes more complex. The more bits that we reserve for compression, the fewer bits we have remaining to construct the best error-correcting code and vice versa. Balancing the two sources of distortion through the optimal bit allocation, in addition to designing the best source and channel codes, makes this *joint source-channel coding* (JSCC) problem challenging.

In addition to these design challenges, real-world communication systems face computational and memory constraints that hinder the straightforward application of Shannon's results. Specifically, practical decoding algorithms rely on approximations that yield suboptimal reconstruction accuracies. The information theory community has studied this problem extensively, and proposed different bounds for finite bit-length JSCC in a wide variety of contexts (Pilc (1967), Csiszar (1982), Kostina & Verdú (2013)). Rather than relying on hand-crafted coding schemes that may require additional tuning in practice, we propose to *learn* the appropriate bit allocations and coding mechanisms using a flexible deep learning framework.

## 3 NEURAL SOURCE AND CHANNEL CODES

Given the above considerations, we explore a learning-based approach to the JSCC problem. To do so, we consider a flexible *class* of codes parameterized by a neural network and jointly optimize for the encoding and decoding procedure. This approach is inspired by recent successes in training (discrete) latent variable models, such as VAEs (Neal (1992), Rolfe (2016), van den Oord et al. (2017)). We explore the connection to different types of autoencoders in section 4.1.

### 3.1 CODING PROCESS

Let $X, \hat{Y}, Y, \hat{X}$ be random variables denoting the inputs, codewords, noisy codewords, and reconstructions respectively. We model their joint distribution $p(x, \hat{y}, y, \hat{x})$ using the following graphical model $X \rightarrow \hat{Y} \rightarrow Y \rightarrow \hat{X}$ as:

$$p(x, \hat{y}, y, \hat{x}) = p_{\text{data}}(x) q_{\text{encoder}}(\hat{y}|x; \phi) p_{\text{channel}}(y|\hat{y}; \epsilon) p_{\text{decoder}}(\hat{x}|y; \theta) \tag{1}$$

In equation 1, $p_{\text{data}}(x)$ denotes the distribution over inputs $\mathcal{X}$. It is not known explicitly in practice, and only accessible through samples. $p_{\text{channel}}(y|\hat{y}; \epsilon)$ is the channel model, where we specifically focus on the binary symmetric channel (BSC) case. The BSC independently flips each bit in the codeword with probability $\epsilon$ (e.g., $0 \rightarrow 1$). Therefore, $\hat{Y}$ takes values in $\hat{\mathcal{Y}} = \mathcal{Y} = \{0, 1\}^m$ and

$$p_{\text{channel}}(y|\hat{y}; \epsilon) = \prod_{i=1}^{m} \epsilon^{y_i \oplus \hat{y}_i} (1 - \epsilon)^{y_i \oplus \hat{y}_i \oplus 1}$$

where $\oplus$ denotes addition modulo 2 (i.e., an eXclusive OR).

A *stochastic* encoder $q_{\text{encoder}}(\hat{y}|x; \phi)$ generates a codeword $\hat{y}$ given an input $x$. Specifically, we model each bit $\hat{y}_i$ in the code with an independent Bernoulli random vector. We model the parameters of this Bernoulli with a neural network $f_\phi(\cdot)$ (an MLP or a CNN) parameterized by $\phi$:

$$q_{\text{encoder}}(\hat{y}|x, \phi) = \prod_{i=1}^{m} \sigma(f_\phi(x_i))^{\hat{y}_i} (1 - \sigma(f_\phi(x_i)))^{(1-\hat{y}_i)}$$

where $\sigma(z)$ denotes the sigmoid function.

Similarly, we posit a probabilistic decoder $p_{\text{decoder}}(\hat{x}|y; \theta)$ parameterized by an MLP/CNN that, given $y$, generates a decoded image $\hat{x}$. We model each pixel as a factorized Gaussian with a fixed, isotropic covariance to yield: $\hat{x}|y \sim \mathcal{N}(f_\theta(y), \sigma^2 I)$, where $f_\theta(\cdot)$ denotes the decoder network.

## 4 END-TO-END VARIATIONAL LEARNING OF NEURAL CODES

In order to learn an effective coding scheme, we **maximize the mutual information** between the input $X$ and the corresponding *noisy* codeword $Y$ (Barber & Agakov (2006)). That is, the code $\hat{y}$ should be robust to partial corruption; even its noisy instantiation $y$ should preserve as much information about the original input $x$ as possible (MacKay (2003)).

First, we note that we can analytically compute the encoding distribution $q_{\text{noisy\_enc}}(y|x; \epsilon, \phi)$ *after* it has been perturbed by the channel by marginalizing over $\hat{y}$:

$$q_{\text{noisy\_enc}}(y|x; \epsilon, \phi) = \sum_{\hat{y} \in \hat{\mathcal{Y}}} q_{\text{encoder}}(\hat{y}|x; \phi) p_{\text{channel}}(y|\hat{y}; \epsilon)$$

which yields:

$$q_{\text{noisy\_enc}}(y|x; \phi, \epsilon) = \prod_{i=1}^{m} \left( \sigma(f_\phi(x_i)) - 2\sigma(f_\phi(x_i))\epsilon + \epsilon \right)^{y_i} \left( 1 - \sigma(f_\phi(x_i)) + 2\sigma(f_\phi(x_i))\epsilon - \epsilon \right)^{(1-y_i)}$$

It is relatively straightforward to handle other types of channel noise; we provide additional examples of the binary erasure channel (BEC) and deletion channel in the Appendix.

We get the following optimization problem:

$$\max_\phi I(X, Y; \phi, \epsilon) = H(X) - H(X|Y; \phi, \epsilon)$$

$$= \mathbb{E}_{x \sim p_{\text{data}}(x)} \mathbb{E}_{y \sim q_{\text{noisy\_enc}}(y|x; \epsilon, \phi)} \left[ \log p(x|y; \epsilon, \phi) \right] + \text{const.} \tag{2}$$

$$\geq \mathbb{E}_{x \sim p_{\text{data}}(x)} \mathbb{E}_{y \sim q_{\text{noisy\_enc}}(y|x; \epsilon, \phi)} \left[ \log p_{\text{decoder}}(x|y; \theta) \right] + \text{const.} \tag{3}$$

where $p(x|y; \epsilon, \phi)$ is the true (and intractable) posterior from Eq. 1 and $p_{\text{decoder}}(x|y; \theta)$ is an amortized variational approximation. The true posterior $p(x|y; \epsilon, \phi)$ — the posterior probability over

possible inputs $x$ given the received noisy codeword $y$ — is the best possible decoder. However, it is also often intractable to evaluate and optimize. We therefore use a *tractable* variational approximation $p_{\text{decoder}}(\hat{x}|y;\theta)$. Crucially, this variational approximation is *amortized* (Kingma & Welling (2013)), and is the inference distribution that will actually be used for decoding at test time. Because of amortization, *decoding is guaranteed to be efficient*, in contrast with existing error correcting codes, which typically involve NP-hard MPE inference queries.

Given any encoder ($\phi$), we can find the best amortized variational approximation by maximizing the lower bound (3) as a function of $\theta$. Therefore, the NECST training objective is given by:

$$\max_{\theta,\phi} \mathbb{E}_{x \sim p_{\text{data}}(x)} \mathbb{E}_{y \sim q_{\text{noisy\_enc}}(y|x;\epsilon,\phi)} \left[ \log p_{\text{decoder}}(x|y;\theta) \right]$$

In practice, we approximate the expectation of the data distribution $p_{\text{data}}(x)$ with a finite dataset $\mathcal{D}$:

$$\max_{\theta,\phi} \sum_{x \in \mathcal{D}} \mathbb{E}_{y \sim q_{\text{noisy\_enc}}(y|x;\epsilon,\phi)} \left[ \log p_{\text{decoder}}(x|y;\theta) \right] \equiv \mathcal{L}(\phi,\theta;x,\epsilon) \tag{4}$$

Thus we can jointly learn the encoding and decoding scheme by optimizing the parameters $\phi$ and $\theta$. In this way, the encoder ($\phi$) is "aware" of the computational limitations of the decoder ($\theta$), and is optimized accordingly. However, a main learning challenge is that we are not able to backpropagate directly through the *discrete* latent variable $y$; we elaborate upon this point further in Section 5.1.

The solution will depend on the number of available bits $m$, the noise level $\epsilon$, and the structure of the input data $x$. For intuition, consider the case $m = 0$. In this case, no information can be transmitted, and the best decoder will fit a single Gaussian to the data. When $m = 1$ and $\epsilon = 0$ (noiseless case), the model will learn a mixture of $2^m = 2$ Gaussians. However, if $m = 1$ and $\epsilon = 0.5$, again no information can be transmitted, and the best decoder will fit a single Gaussian to the data. Adding noise forces the model to decide how it should effectively partition the data such that (1) similar items are grouped together in the same cluster; and (2) the clusters are "well-separated" (even when a few bits are flipped, they do not "cross over" to a different cluster that will be decoded incorrectly). Thus, from the perspective of unsupervised learning, NECST attempts to learn *robust binary representations* of the data.

## 4.1 NECST as a Generative Model

The objective function in equation (4) closely resembles those commonly used in generative modeling frameworks; in fact, NECST can also be viewed as a generative model. In its simplest form, our model with a noiseless channel (i.e., $\epsilon = 0$), deterministic encoder, and deterministic decoder is identical to a traditional autoencoder (Bengio et al. (2007)). Once channel noise is present ($\epsilon > 0$) and the encoder/decoder become probabilistic, NECST begins to more closely resemble other variational autoencoding frameworks. Specifically, NECST is similar to Denoising Autoencoders (DAE) (Vincent et al. (2008)), except that it is explicitly trained for robustness to partial destruction of *the latent space*, as opposed to the input space.

NECST is also a variant of the VAE, with two nuanced distinctions. While both models learn a joint distribution over the observations and latent codes, the VAE: (1) optimizes a variational lower bound to *the marginal log-likelihood* $p(X)$ as opposed to the mutual information $I(X, Y)$, and (2) explicitly posits a prior distribution $p(Y)$ over the latent variables. Their close relationship is evidenced by a line of work on rate-distortion optimization in the context of VAEs (Ballé et al. (2016), Alemi et al. (2017)), as well as other information-theoretic interpretations of the VAE's information preference (Hinton & Van Camp (1993), Honkela & Valpola (2004), Chen et al. (2016)). We also note that existing autoencoders are aimed at compression, while for sufficiently large $m$ NECST will attempt to learn lossless compression with added redundancy for error correction.

Finally, NECST can be seen as a discrete version of the Uncertainty Autoencoder (UAE) (Grover & Ermon (2018)). The two models share identical objective functions with two notable differences: For NECST, (1) the latent codes $Y$ are discrete random variables, and (2) the noise model is no longer continuous. The special properties of the continuous UAE carry over directly to its discrete counterpart. Grover & Ermon (2018) proved that under certain conditions, the UAE specifies an implicit generative model (Diggle & Gratton (1984), Mohamed & Lakshminarayanan (2016)). We restate their major theorem and extend their results here.

Starting from any data point $x^{(0)} \sim \mathcal{X}$, we define a Markov chain over $\mathcal{X} \times \mathcal{Y}$ with the following transitions:

$$y^{(t)} \sim q_{\text{noisy\_enc}}(y|x^{(t)}; \phi, \epsilon) \;\;,\;\; x^{(t+1)} \sim p_{\text{decoder}}(x|y^{(t)}; \theta) \tag{5}$$

**Theorem 1.** *For any fixed value of $\phi$ and $\epsilon > 0$, suppose that the NECST objective is globally maximized for some choice of $\theta^*$ so that inequality (3) is tight. Then the Markov Chain (5) with parameters $\phi$ and $\theta^*$ is ergodic and its stationary distribution is given by $p_{data}(x)q_{noisy\_enc}(y \mid x; \epsilon, \phi)$.*

*Proof.* See Appendix A.1. □

Hence NECST has an intractable likelihood but can generate samples from $p_{\text{data}}$ by running the chain above. Samples initialized from both random noise and test data can be found in Appendix D.5.

## 5 EXPERIMENTAL RESULTS

We first review the optimization challenges of training discrete latent variable models and elaborate on our training procedure. Then to validate our work, we first assess NECST's compression and error correction capabilities against a combination of two widely-used compression (JPEG, VAE) and channel coding (ideal code, LDPC (Gallager (1962))) algorithms. We experiment on randomly generated length-100 bitstrings, MNIST (LeCun (1998)), Omniglot (Lake et al. (2015)), Street View Housing Numbers (SVHN) (Netzer et al. (2011)), and CelebA (Liu et al. (2015)) datasets to account for different image sizes and colors. Next, we test the decoding speed of NECST's neural decoder and find that it performs upwards of *an order of magnitude faster* than standard decoding algorithms based on iterative belief propagation (and two orders of magnitude on GPU). Finally, we assess the quality of the latent codes after training, and examine interpolations in latent space as well as how well the learned features perform for downstream classification tasks.

### 5.1 OPTIMIZATION PROCEDURE

Recall the NECST objective in equation 4. While obtaining Monte Carlo-based gradient estimates with respect to $\theta$ is easy, gradient estimation with respect to $\phi$ is challenging because these parameters specify the Bernoulli random variable $q_{\text{noisy\_enc}}(y|x; \epsilon, \phi)$. The commonly used *reparameterization trick* cannot be applied in this setting, as the discrete stochastic unit in the computation graph renders the overall network non-differentiable (Schulman et al. (2015)).

A simple alternative is to use the *score function estimator* in place of the gradient, as defined in the REINFORCE algorithm (Williams (1992)). However, this estimator suffers from high variance, and several others have explored different formulations and control variates to mitigate this issue (Wang et al. (2013), Gu et al. (2015), Ruiz et al. (2016), Tucker et al. (2017), Grathwohl et al. (2017)). Others have proposed a continuous relaxation of the discrete random variables, as in the Gumbel-softmax (Jang et al. (2016)) and Concrete (Maddison et al. (2016)) distributions.

To preserve the hard "discreteness" of the latent codes, we used VIMCO (Mnih & Rezende (2016)), a multi-sample variational lower bound objective for obtaining low-variance gradients. VIMCO constructs leave-one-out control variates using its samples, as opposed to the single-sample objective NVIL (Mnih & Gregor (2014)) which requires learning additional baselines during training. Thus, we used the 5-sample VIMCO objective in subsequent experiments for the optimization procedure, leading us to our final multi-sample ($K = 5$) objective:

$$\mathcal{L}^K(\phi, \theta; x, \epsilon) = \max_{\theta, \phi} \sum_{x \in \mathcal{D}} \mathbb{E}_{y^{1:K} \sim q_{\text{noisy\_enc}}(y|x; \epsilon, \phi)} \left[ \log \frac{1}{K} \sum_{i=1}^K p_{\text{decoder}}(x|y^i; \theta) \right] \tag{6}$$

### 5.2 FIXED DISTORTION: JPEG + IDEAL CHANNEL CODE

In this experiment, we compare the performances of: (1) NECST and (2) JPEG + ideal channel code in terms of compression. Specifically, we fix the number of bits $m$ used by NECST to source and channel code and obtain the corresponding distortion levels (reconstruction errors) at various noise levels $\epsilon$. For fixed $m$, distortion will increase with $\epsilon$. Then, for each noise level we estimate the number of bits an alternate system using JPEG and an ideal channel code — the best that is

theoretically possible — would have to use to match NECST's distortion levels. Assuming an ideal channel code implies that all messages will be transmitted perfectly across the noisy channel; thus the resulting distortion will only be a function of the compression mechanism. We note that some approximations commonly used in the information theory community are used to obtain the analytic expressions used for this estimate. Full details and additional results may be found in Appendix D.1.

We find that with the exception of random data (Appendix D.1), NECST excels at compression; Figure 1 shows that in order to achieve the same level of distortion as NECST, the JPEG-ideal channel code system requires a much larger number bits *across all noise levels*. In particular, we note that NECST is slightly more effective than JPEG at pure compression ($\epsilon = 0$), and becomes significantly better at higher noise levels (e.g., for $\epsilon = 0.4$, NECST requires $20\times$ less bits). The negative result for random bits is to be expected because the data has no exploitable statistical dependencies.

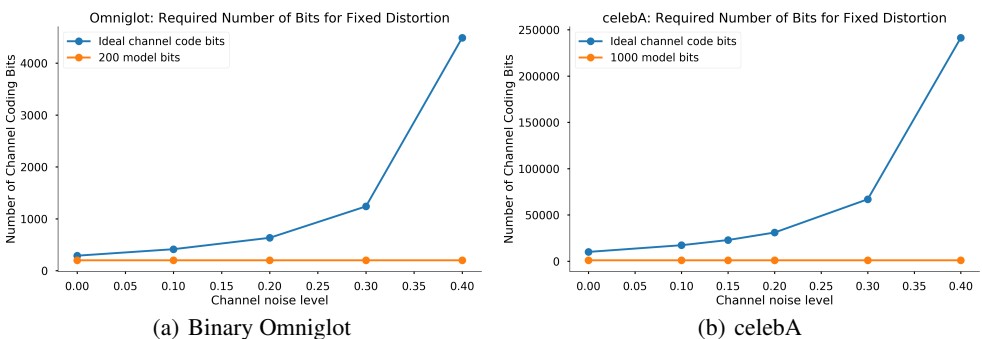

(a) Binary Omniglot                    (b) celebA

Figure 1: Theoretical $m$ required for JPEG + ideal channel code to match NECST's distortion.

### 5.3 FIXED RATE: SOURCE CODING + LDPC

Next, we compare the performances of: (1) NECST and (2) discrete VAE + LDPC in terms of distortion. Specifically, we fix the bit-length budget for both systems and evaluate whether learning compression and error-correction jointly (NECST) improves reconstruction errors (distortion) for the same rate. We do not report a comparison against JPEG + LDPC as we were often unable to obtain valid JPEG files to compute distortions after imperfect LDPC decoding. However, as we have seen in the JPEG + ideal channel code experiment, we can reasonably assume that the JPEG-LDPC system would have displayed worse performance.

We experiment with a discrete VAE with a uniform prior over the latent codes for encoding. To fix the total number of bits that are transmitted through the channel, we double the length of the VAE encodings with an LDPC channel code to match NECST's latent code dimension $m$. We note from Figure 2a that although the resulting distortion is similar, NECST outperforms the VAE across all noise levels for image datasets. These results suggest that NECST's learned coding scheme performs at least as well as LDPCs, an industrial-strength, widely deployed class of error correcting codes.

### 5.4 DECODING TIME

Another advantage of NECST over traditional channel codes is that after training, the amortized decoder can very efficiently map the transmitted code into its best reconstruction at test time. Other sparse graph-based coding schemes such as LDPC, however, are more time-consuming because decoding involves an NP-hard optimization problem typically approximated with multiple iterations of belief propagation.

To compare the speed of our neural decoder to LDPC's belief propagation, we fixed the number of bits that were transmitted across the channel and averaged the total amount of time taken for decoding across ten runs. The results are shown below in Figure 2b for the binarized MNIST and Omniglot datasets, and refer the reader to Appendix D.7 for the simulated random data. On CPU, NECST's decoder averages an order of magnitude faster than the LDPC decoder running for 50 iterations, which is based on an optimized C implementation. On GPU, NECST displays *two orders of magnitude in speedup*.

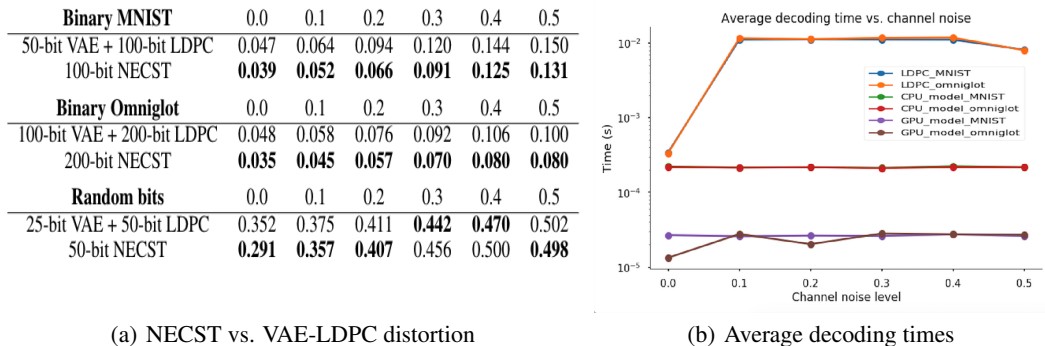

| Binary MNIST | 0.0 | 0.1 | 0.2 | 0.3 | 0.4 | 0.5 |
|---|---|---|---|---|---|---|
| 50-bit VAE + 100-bit LDPC | 0.047 | 0.064 | 0.094 | 0.120 | 0.144 | 0.150 |
| 100-bit NECST | **0.039** | **0.052** | **0.066** | **0.091** | **0.125** | **0.131** |
| **Binary Omniglot** | 0.0 | 0.1 | 0.2 | 0.3 | 0.4 | 0.5 |
| 100-bit VAE + 200-bit LDPC | 0.048 | 0.058 | 0.076 | 0.092 | 0.106 | 0.100 |
| 200-bit NECST | **0.035** | **0.045** | **0.057** | **0.070** | **0.080** | **0.080** |
| **Random bits** | 0.0 | 0.1 | 0.2 | 0.3 | 0.4 | 0.5 |
| 25-bit VAE + 50-bit LDPC | 0.352 | 0.375 | 0.411 | **0.442** | **0.470** | 0.502 |
| 50-bit NECST | **0.291** | **0.357** | **0.407** | 0.456 | 0.500 | **0.498** |

(a) NECST vs. VAE-LDPC distortion

(b) Average decoding times

Figure 2: (Left) Reconstruction error of NECST vs. VAE + rate-1/2 LDPC codes. Lower is better. (Right) Average decoding times for NECST vs. 50 iterations of LDPC decoding.

## 6 NECST FOR ROBUST REPRESENTATION LEARNING

In addition to its ability to source and channel code effectively, NECST also serves as an implicit generative model that yields robust and interpretable latent representations and realistic samples from the underlying data distribution. Specifically, in light of Theorem 1, we can think of $q_{\text{encoder}}(\widehat{y}|x; \phi)$ as mapping images $x$ to latent representations $\widehat{y}$.

### 6.1 INTERPOLATION IN LATENT SPACE

To assess whether the model has: (1) injected redundancies into the learned codes and (2) learned interesting features, we interpolate between different data points in latent space and qualitatively observe whether the model captures semantically meaningful variations in the data. We select two test points to be the start and end, sequentially flip one bit at a time in the latent code, and pass the altered code through the decoder to observe how the reconstruction changes. In Figure 3, we show two illustrative examples from the MNIST digits. From the starting digit, each bit-flip slowly alters characteristic features such as rotation, thickness, and stroke style until the digit is reconstructed to something else entirely. We note that because NECST is trained to encode redundancies, we do not observe a drastic change per flip. Rather, it takes a significant level of corruption for the decoder to interpret the original digit as another. Also, due to the i.i.d. nature of the channel noise model, we observe that the sequence at which the bits are flipped do not have a significant effect on the reconstructions. Additional interpolations may be found in Appedix D.4.

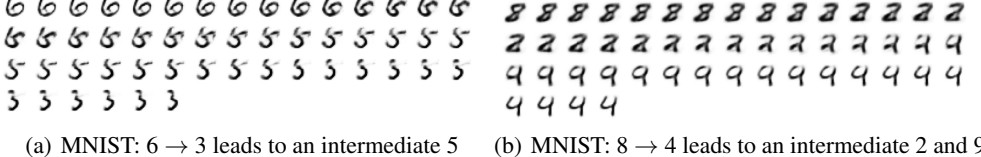

(a) MNIST: 6 → 3 leads to an intermediate 5    (b) MNIST: 8 → 4 leads to an intermediate 2 and 9

Figure 3: Latent space interpolation, where each image is obtained by flipping one additional latent bit and passing through the decoder.

### 6.2 DOWNSTREAM CLASSIFICATION

To demonstrate that the latent representations learned by NECST are useful for downstream classification tasks, we extract the binary features and train eight different classification algorithms: $k$-nearest neighbors (KNN), decision trees (DT), random forests (RF), multilayer perceptron (MLP), AdaBoost (AdaB), Quadratic Discriminant Analysis (QDA), and support vector machines (SVM). As shown on the leftmost numbers in Table 1, the simple classifiers perform reasonably well in the digit classification task for MNIST ($\epsilon = 0$). We observe that with the exception of KNN, all other

classifiers achieve higher levels of accuracy when using features that were trained with simulated channel noise ($\epsilon = 0.1, 0.2$).

Table 1: Classification accuracy on MNIST/noisy MNIST using 100-bit features from NECST.

| Noise $\epsilon$ | KNN | DT | RF | MLP | AdaB | NB | QDA | SVM |
|---|---|---|---|---|---|---|---|---|
| 0 | **0.95/0.86** | 0.65/0.54 | 0.71/0.59 | 0.93/0.87 | 0.72/0.65 | 0.75/0.65 | 0.56/0.28 | 0.88/0.81 |
| 0.1 | 0.95/0.86 | 0.65/0.59 | 0.74/0.65 | **0.934**/0.88 | 0.74/0.72 | 0.83/0.77 | **0.94/0.90** | 0.92/0.84 |
| 0.2 | 0.94/0.86 | **0.78/0.69** | **0.81/0.76** | 0.93/**0.89** | **0.78/0.80** | **0.87/0.81** | 0.93/0.90 | **0.93/0.86** |

To further test the hypothesis that NECST trained with the noisy channel learns more robust latent features, we freeze the pre-trained model and evaluate classification accuracy using a "noisy" MNIST dataset. We synthesized noisy MNIST by adding $\epsilon \sim \mathcal{N}(0, 0.5)$ noise to all pixel values in MNIST, and ran the same experiment as above. As shown in the rightmost numbers of Table 1, we again observe that most algorithms show improved performance with added channel noise.

# 7 RELATED WORK

There has been a recent surge of work applying deep learning and generative modeling techniques to lossy image compression, many of which compare favorably against industry standards such as JPEG, JPEG2000, and WebP (Toderici et al. (2015), Ballé et al. (2016), Toderici et al. (2017)). Theis et al. (2017) use compressive autoencoders that learn the optimal number of bits to represent images based on their pixel frequencies. Ballé et al. (2018) use a variational autoencoder (VAE) (Kingma & Welling (2013) Rezende et al. (2014)) with a learnable scale hyperprior to capture the image's partition structure as side information for more efficient compression. Santurkar et al. (2017) use adversarial training (Goodfellow et al. (2014)) to learn neural codecs for compressing images and videos using DCGAN-style ConvNets (Radford et al. (2015)). Yet these methods focus on source coding only, and do not consider the setting where the compression must be robust to channel noise.

In a similar vein, there has been growing interest on leveraging these deep learning systems to sidestep the use of hand-designed codes. Several lines of work train neural decoders based on known coding schemes, sometimes learning more general decoding algorithms than before (Nachmani et al. (2016), Gruber et al. (2017), Cammerer et al. (2017), Dorner et al. (2017)). (Kim et al. (2018a), Kim et al. (2018b)) parameterize sequential codes with recurrent neural network (RNN) architectures that achieve comparable performance to capacity-approaching codes over the additive white noise Gaussian (AWGN) and bursty noise channels. However, source coding is out of the scope for these methods that focus on learning good channel codes.

The problem of end-to-end transmission of structured data, on the other hand, is less well-studied. Farsad et al. (2018) use RNNs to communicate text over a discrete, binary erasure channel (BEC) and are able to preserve the words' semantics. The most similar to our work is that of Bourtsoulatze et al. (2018), who use autoencoders for transmitting images over the AWGN and slow Rayleigh fading channels, which are continuous. We provide a holistic treatment of various discrete noise models and show how NECST can also be used for unsupervised learning.

# 8 DISCUSSION

We described how NECST can be used to learn an efficient joint source-channel coding scheme by simulating a noisy channel during training. We showed that the model: (1) is competitive against a combination of industry standard compression and error-correcting codes, (2) learns an extremely fast neural decoder through amortized inference, and (3) learns a latent code that is not only robust to corruption, but also useful for general downstream tasks such as classification.

One limitation of NECST is that we need to train the model separately for different code-lengths and datasets; in its current form, it can only handle fixed-length codes. Extending the model to streaming or adaptive learning scenarios that allow for learning variable-length codes is an exciting direction for future work. Another direction would be to analyze the characteristics and properties of the learned latent codes under different discrete channel models, such as erasure channels.

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

APPENDIX

## A    PROOFS OF THEORETICAL RESULTS

### A.1    STATIONARY DISTRIBUTION OF MARKOV CHAIN

We follow the proof structure in (Grover & Ermon (2018)) with minor adaptations. First, we note that the Markov chain defined in Eq. 5 is ergodic due to the BSC noise model. This is because the BSC defines a distribution over all possible (finite) configurations of the latent code $\mathcal{Y}$, and the Gaussian decoder posits a non-negative probability of transitioning to all possible reconstructions $\mathcal{X}$. Thus given any $(X, Y)$ and $(\mathcal{X}', \mathcal{Y}')$ such that the density $p(\mathcal{X}, \mathcal{Y}) > 0$ and $p(\mathcal{X}', \mathcal{Y}') > 0$, the probability density of transitioning $p(\mathcal{X}', \mathcal{Y}'|\mathcal{X}, \mathcal{Y}) > 0$.

Next, we can rewrite the objective in Eq. equation 4 as the following:

$$\mathbb{E}_{p(x,y;\epsilon,\phi)}\left[\log p_{\text{decoder}}(x|y;\theta)\right] = \mathbb{E}_{q(y;,\phi)}\left[\int p(x|y;\phi)\log p_{\text{decoder}}(x|y;\theta)\right]$$
$$= -H(X|Y;\phi) - \mathbb{E}_{q(y;\epsilon,\phi)}[KL(p(X|y;\phi)||p(X|y;\theta))]$$

We note that for any value of $\phi$, the optimal value of $\theta = \theta^*$ eliminates the KL-divergence term, as it is non-negative and minimized only when its argument distributions are identical. Then, for any $\phi$, we note that the following Gibbs chain converges to $p(x, y)$ if the chain is ergodic:

$$y^{(t)} \sim q_{\text{noisy\_enc}}(y|x^{(t)}; \phi, \epsilon) \ , \ x^{(t+1)} \sim p(x|y^{(t)}; \theta^*) \tag{7}$$

Substituting $p(x|y; \theta^*)$ for $p_{\text{decoder}}(x|y; \theta)$ finishes the proof.

## B    ADDITIONAL CHANNEL MODELS

### B.1    BINARY ERASURE CHANNEL (BEC)

The BEC erases each bit at position $i$ in the latent code $\hat{y}$ with some i.i.d. probability $\epsilon$ into a new symbol. Thus if the input to the channel is $\hat{y}|x \sim \text{Bern}(\sigma(f_\phi(x)))$, the BEC induces a 3-way Categorical distribution where the third category refers to the bit-erasure. We denote the erased bit as ?, which can be modeled as category 2 in the new alphabet of $\mathcal{Y} = \{0, 1, 2\}^m$. This yields:

$$y|x \sim \text{Cat}(1 - \epsilon - \sigma(f_\phi(x)) + \sigma(f_\phi(x)) \cdot \epsilon, \sigma(f_\phi(x)) - \sigma(f_\phi(x)) \cdot \epsilon, \epsilon)$$

where:
$$p(y = 0|\hat{y} = 0, x) = 1 - \epsilon - \sigma(f_\phi(x)) + \sigma(f_\phi(x)) \cdot \epsilon$$
$$p(y = 1|\hat{y} = 1, x) = \sigma(f_\phi(x)) - \sigma(f_\phi(x)) \cdot \epsilon$$
$$p(y = ?|\hat{y} = 1, x) + p(y = ?|\hat{y} = 0, x) = \epsilon$$

### B.2    DELETION CHANNEL

The deletion channel is similar to the BEC, except that with some independent probability $\epsilon$, the bit at position $i$ is deleted. Thus our codewords $y$ may not necessarily be of length $m$ as in $\hat{y} \in \{0, 1\}^m$, and the space of $y$ may have variable lengths.

## C    NECST ARCHITECTURE AND HYPERPARAMETER CONFIGURATIONS

### C.1    MNIST

For MNIST, we used the static binarized version as provided in (Burda et al. (2015)) with train/validation/test splits of 50K/10K/10K respectively.

- encoder: MLP with 1 hidden layer (500 hidden units), ReLU activations

- decoder: 2-layer MLP with 500 hidden units each, ReLU activations. The final output layer has a sigmoid activation for learning the parameters of $p_{\text{noisy\_enc}}(y|x; \phi, \epsilon)$
- n_bits: 100
- n_epochs: 200
- batch size: 100
- L2 regularization penalty of encoder weights: 0.001
- Adam optimizer with lr=0.001

## C.2 OMNIGLOT

We statically binarize the Omniglot dataset by rounding values above 0.5 to 1, and those below to 0.

- encoder: MLP with 1 hidden layer (500 hidden units), ReLU activations
- decoder: 2-layer MLP with 500 hidden units each, ReLU activations. The final output layer has a sigmoid activation for learning the parameters of $p_{\text{noisy\_enc}}(y|x; \phi, \epsilon)$
- n_bits: 200
- n_epochs: 500
- batch size: 100
- L2 regularization penalty of encoder weights: 0.001
- Adam optimizer with lr=0.001

## C.3 RANDOM BITS

We randomly generated length-100 bitstrings by drawing from a Bern(0.5) distribution for each entry in the bitstring. The train/validation/test splits are: 5K/1K/1K.

- encoder: MLP with 1 hidden layer (500 hidden units), ReLU activations
- decoder: 2-layer MLP with 500 hidden units each, ReLU activations. The final output layer has a sigmoid activation for learning the parameters of $p_{\text{noisy\_enc}}(y|x; \phi, \epsilon)$
- n_bits: 50
- n_epochs: 200
- batch size: 100
- L2 regularization penalty of encoder weights: 0.001
- Adam optimizer with lr=0.001

## C.4 SVHN

For SVHN, we collapse the "easier" additional examples with the more difficult training set, and randomly partition 10K of the roughly 600K dataset into a validation set.

- encoder: CNN with 3 convolutional layers + fc layer, ReLU activations
- decoder: CNN with 4 deconvolutional layers, ReLU activations.
- n_bits: 500
- n_epochs: 500
- batch size: 100
- L2 regularization penalty of encoder weights: 0.001
- Adam optimizer with lr=0.001

The CNN architecture for the encoder is as follows:

1. conv1 = n_filters=128, kernel_size=2, strides=2, padding="VALID"

2. conv2 = n_filters=256, kernel_size=2, strides=2, padding="VALID"

3. conv3 = n_filters=512, kernel_size=2, strides=2, padding="VALID"

4. fc = 4*4*512 → n_bits, no activation

The decoder architecture follows the reverse, but without a final deconvolution layer as: n_filters=3, kernel_size=1, strides=1, padding="VALID", activation=ReLU.

### C.5   CELEBA

We use the celebA dataset with standard train/validation/test splits with minor preprocessing. First, we align and crop each image to focus on the face, resizing the image to be $(64, 64, 3)$.

- encoder: CNN with 5 convolutional layers + fc layer, ELU activations
- decoder: CNN with 5 deconvolutional layers, ELU activations.
- n_bits: 1000
- n_epochs: 500
- batch size: 100
- L2 regularization penalty of encoder weights: 0.001
- Adam optimizer with lr=0.0001

The CNN architecture for the encoder is as follows:

1. conv1 = n_filters=32, kernel_size=4, strides=2, padding="SAME"

2. conv2 = n_filters=32, kernel_size=4, strides=2, padding="SAME"

3. conv3 = n_filters=64, kernel_size=4, strides=2, padding="SAME"

4. conv4 = n_filters=64, kernel_size=4, strides=2, padding="SAME"

5. conv5 = n_filters=256, kernel_size=4, strides=2, padding="VALID"

6. fc = 256 → n_bits, no activation

The decoder architecture follows the reverse, but without the final fully connected layer and the last deconvolutional layer as: n_filters=3, kernel_size=4, strides=2, padding="SAME", activation=sigmoid.

## D   ADDITIONAL EXPERIMENTAL DETAILS AND RESULTS

### D.1   FIXED DISTORTION: JPEG-IDEAL CHANNEL CODE SYSTEM

For the BSC channel, we can compute the theoretical channel capacity with the formula $C = 1 - H_b(\epsilon)$, where $\epsilon$ denotes the bit-flip probability of the channel and $H_b$ denotes the binary entropy function. Note that the communication rate of $C$ is achievable in the asymptotic scenario of infinitely long messages; in the finite bit-length regime, particularly in the case of short blocklengths, the highest achievable rate will be much lower.

For each image, we first obtain the target distortion $d$ per channel noise level by using a fixed bit-length budget with NECST. Next we use the JPEG compressor to encode the image at the distortion level $d$. The resulting size of the compressed image $f(d)$ is used to get an estimate $f(d)/C$ for the number of bits used for the image representation in the ideal channel code scenario. While measuring the compressed image size $f(d)$, we ignore the header size of the JPEG image, as the header is similar for images from the same dataset.

The plots compare $f(d)/C$ with $m$, the fixed bit-length budget for NECST.

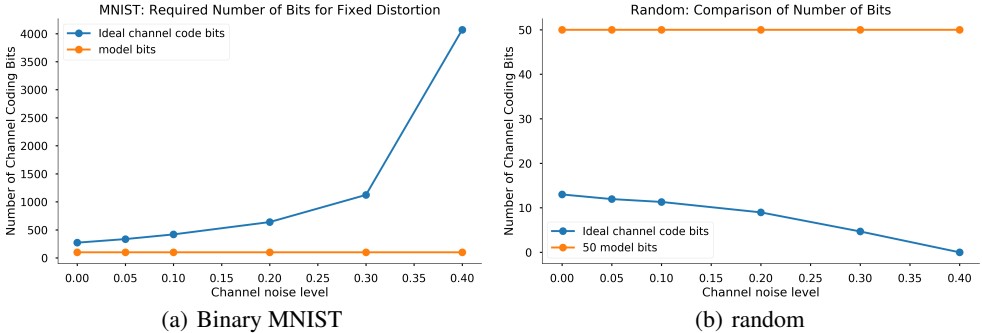

(a) Binary MNIST            (b) random

Figure 4: Theoretical $m$ required for JPEG + ideal channel code to match NECST's distortion.

## D.2 Fixed Rate: JPEG-LDPC system

We first elaborate on the usage of the LDPC software. We use an optimized C implementation of LDPC codes to run our decoding time experiments: (http://radfordneal.github.io/LDPC-codes/).

The procedure is as follows:

- `make-pchk` to create a parity check matrix for a regular LDPC code with three 1's per column, eliminating cycles of length 4 (default setting). The number of parity check bits is: `total number of bits allowed - length of codeword`.
- `make-gen` to create the generator matrix from the parity check matrix. We use the default `dense` setting to create a dense representation.
- `encode` to encode the source bits into the LDPC encoding
- `transmit` to transmit the LDPC code through a `bsc` channel, with the appropriate level of channel noise specified (e.g. 0.1)
- `extract` to obtain the actual decoded bits, or `decode` to directly obtain the bit errors from the source to the decoding.

LDPC-based channel codes require larger blocklengths to be effective. To perform an end-to-end experiment with the JPEG compression and LDPC channel codes, we form the input by concatenating multiple blocks of images together into a grid-like image. In the first step, the fixed rate of $m$ is scaled by the total number of images combined, and this quantity is used to estimate the target $f(d)$ to which we compress the concatenated images. In the second step, the compressed concatenated image is coded together by the LDPC code into a bit-string, so as to correct for any errors due to channel noise.

Finally, we decode the corrupted bit-string using the LDPC decoder. The plots compare the resulting distortion of the compressed concatenated block of images with the average distortion on compressing the images individually using NECST. Note that, the experiment gives a slight disadvantage to NECST as it compresses every image individually, while JPEG compresses multiple images together.

We report the average distortions for sampled images from the test set.

Unfortunately, we were unable to run the end-to-end for some scenarios and samples due to errors in the decoding (LDPC decoding, invalid JPEGs etc.).

## D.3 VAE-LDPC system

For the VAE-LDPC system, we place a uniform prior over all the possible latent codes and compute the KL penalty term between this prior $p(y)$ and the random variable $q_{\text{noisy\_enc}}(y|x; \phi, \epsilon)$. The learning rate, batch size, and choice of architecture are data-dependent and fixed to match those of NECST as outlined in Section C. However, we use half the number of bits as allotted for NECST so

that during LDPC channel coding, we can double the codeword length in order to match the rate of our model.

## D.4   INTERPOLATION IN LATENT SPACE

We show results from latent space interpolation for two additional datasets: SVHN and celebA. We used 500 bits for SVHN and 1000 bits for celebA across channel noise levels of [0.0, 0.1, 0.2, 0.3, 0.4, 0.5].

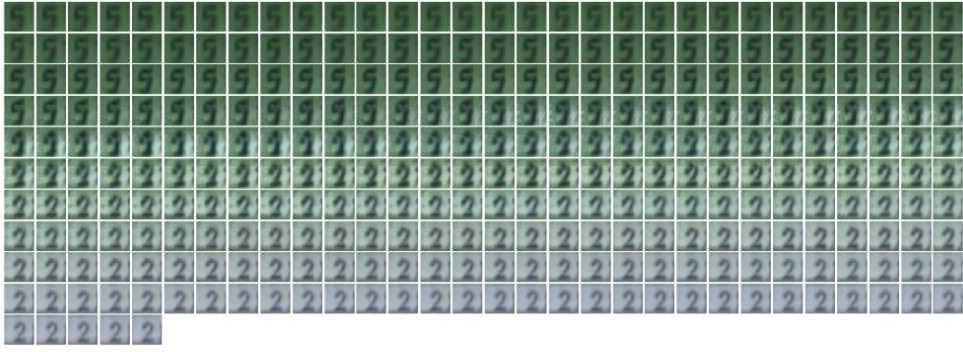

Figure 5: Latent space interpolation of $5 \rightarrow 2$ for SVHN, 500 bits at noise=0.1

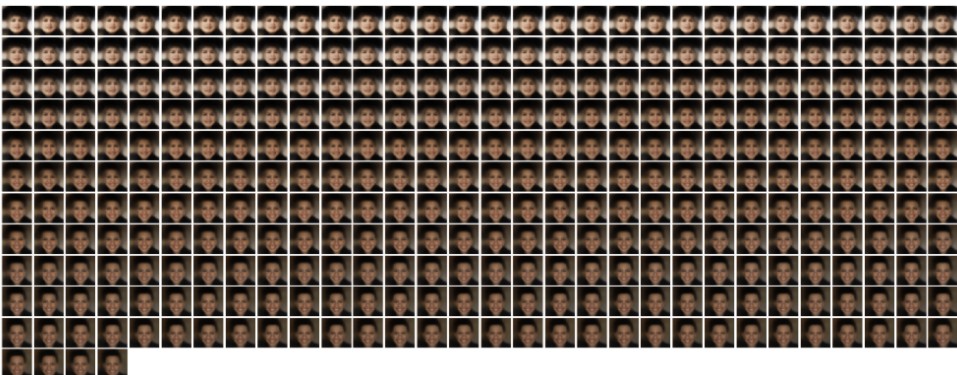

Figure 6: Latent space interpolation for celebA, 1000 bits at noise=0.1

## D.5   MARKOV CHAIN IMAGE GENERATION

We observe that we can generate diverse samples from the data distribution after initializing the chain with both: (1) examples from the test set and (2) random Gaussian noise $x_0 \sim \mathcal{N}(0, 0.5)$.

## D.6   DOWNSTREAM CLASSIFICATION

Following the setup of (Grover & Ermon (2018)), we used standard implementations in sklearn with default parameters for all 8 classifiers with the following exceptions:

1. KNN: n_neighbors=3
2. DF: max_depth=5
3. RF: max_depth=5, n_estimators=10, max_features=1
4. MLP: alpha=1
5. SVC: kernel=linear, C=0.025

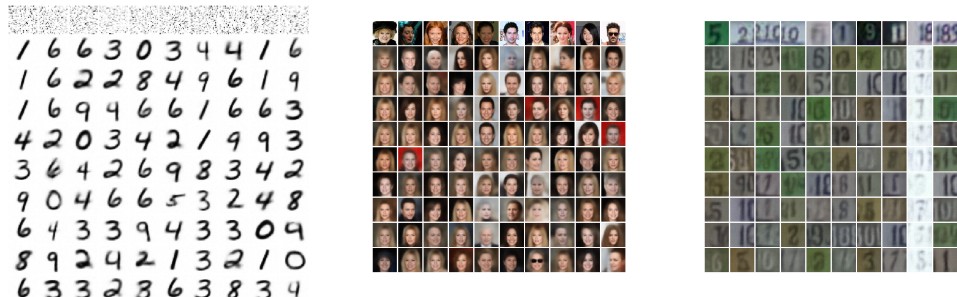

(a) MNIST, initialized from random noise   (b) celebA, initialized from data   (c) SVHN, initialized from data

Figure 7: Markov chain image generation after 9000 timesteps, sampled per 1000 steps

## D.7 DECODING TIME

We show the average decoding times for the randomly generated bitstring data, averaged over 10 runs:

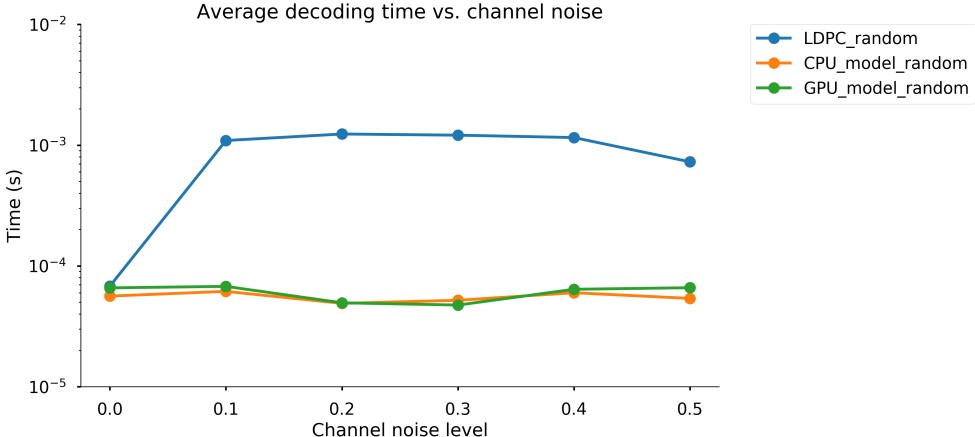

Figure 8: Average decoding times for NECST vs. 50 iterations of BP (LDPC decoding)

