# OpenReview forum: "NECST: Neural Joint Source-Channel Coding"
_ICLR.cc/2019/Conference_

### Official Review · AnonReviewer3 · 2018-11-02
**Good paper, well written and well motivated, good results, no-source code!**

**Rating:** 7
**Confidence:** 4

**Review:**

This interesting paper tackles the problem of joint source-channel coding, by means of learning.

From 100kft heights, especially given the choice of VIMCO gradient estimates, this is effectively a "let's embed a source-channel-decoder simulator and differentiate through it", and find a solution that is better than source|channel factorized classic methods, or hand-tuned approaches.

The method and results are good. The authors also show some interesting results about the representations learned, about how decoded samples (images) change smoothly when the (discrete) embedding (the-codes) changes over deltas of hamming_d()=1bit. This is very good results IHMO. One limitation of this method is the fixed-code-length.

Jumping straight to my main main issue with this paper: no code was made available, at least not at this time.

While the authors do provide an extensive appendix with hyper-parameter specs, usually in my experience when dealing with discrete / monte-carlo methods, it's usually rather hard to reproduce results. I really strongly advise the authors to provide fully reproducible code for this paper, to help further research on this topic.

Besides that I have three technical comments / request regarding this paper:

1// the choice of BSC channel - while this is the easiest most natural choice, and we should certainly have results on BSC, I am left wondering why the authors didn't try other more complex / more realistic channels? The authors only mention this as potential area of future research in the last sentence of the conclusions.

There are several reasons for this comment: first of all, it is well known that even classic joint source-channel coding methods do shine on complex channels, such fading/erasure channels and/or in general channels with correlated error sequences. Such channels are indeed key in modern wireless communications, and are easy to simulate. Given that more-complex channels could be introduced in the channel model p(y_hat|y) -  it would not change the rest of the method - it would be particularly interesting to see what results this method achieve in these more complex environments.

2// I would like to hear more about the choice of VIMCO. Understood the authors statement to "preserve the hard discreteness" ~ that said methods like Gumbel-SM and several others also referenced in the paper ~ have been used  successfully to solve for propagating gradients through discrete units. This is where, in my opinion, experiments comparing VIMCO approximation results to at least one other method could allow to decide / validate the best architecture.

This is also because, in my previous experience, this type of networks with discrete units may be hard to train. I would like to hear from the authors about how stable the training was under different hyper-parameters, and perhaps see some convergence curves for the loss function(s).

3// it's not 100% clear to me where the limitation of fixed code-length come into play from the architecture. Could the authors please point this out clearly?

Thank you!

---

> ### Author Response · Authors · 2018-11-18
> **justifications and clarifications**
>
> We thank the reviewer for their insightful comments and feedback regarding the paper! We address the main points of the reviewer’s concern below:
>
> 1. No source code: We agree with the reviewer that providing code will facilitate future research, and will make the source code publicly available.
>
> 2. More complex channels: We agree with the reviewer that simulating different channels (e.g. fading/erasure/correlated error sequences) would certainly be very interesting, but we note that the BSC is a more difficult channel to work with than the erasure channel. It’d be especially interesting to see the effect on the “features” learned by the model under different noise models.
>
> 3. Choice of VIMCO: We agree with the reviewer in that experiments comparing VIMCO to other methods of training discrete latents would be useful. Our decision to use VIMCO was motivated by earlier experiments using Gumbel-Softmax. As mentioned in our reply to Reviewer 3, we found that using a continuous relaxation of the discrete latent variables allowed the latent codes to capture more information than should be necessary during training. Then at validation/test time, the latent codes would be forced to be discrete. This led to worse reconstructions and samples overall.
>
> 4. Stability of hyperparameters: We did not have too much difficulty across different hyperparameters. The biggest issue we ran into when training on more complex datasets (e.g. celebA) was that we had to decrease the learning rate by an order of magnitude as compared to a simpler dataset. The more complex datasets such as celebA required more complex architectures as well -- the quality of reconstructions improved drastically when we used a convolutional architecture as opposed to an MLP.
>
> 5. Fixed code length limitation: The NECST architecture requires the user to pre-specify the allotted bit-length budget N for learning the latent codes. Therefore, the model is only able to encode images in a particular training set to that fixed N, and cannot encode to codes that are shorter or longer than N. One could imagine that in the case of entropy coding, it may be more efficient to encode frequently occurring images using a shorter-length code, while encoding less common images using longer codes. In its current form, NECST is not able to adaptively learn different code lengths over a given dataset. We plan to address this limitation in follow up work.

---

> > ### Comment · AnonReviewer3 · 2018-11-19
> > **thank you for the reply**
> >
> > Dear authors ~ thank you for the reply ~ and for agreeing to make the source code available.
> >
> > We are pretty much in synch here and now, so I stand by my ACCEPT rating.
> >
> > Regards!
> >
> > .<AnonReviewer3>

---

### Official Review · AnonReviewer2 · 2018-11-04

**Rating:** 4
**Confidence:** 3

**Review:**

Summary of paper: For the finite-bit case of the noisy communication channel model, it is suboptimal to optimize source coding (compression of input) and error correction (fault tolerance for inherent noise in the channel) separately. The authors propose a neural network model (NECST) that is very similar to the standard VAE, except using binary latents with corruption (e.g.,  random bit flipping in the style of a binary symmetric channel). They use VIMCO to optimize through the discrete units. In their experiments, they show that they can outperform a JPEG+ideal channel code model, but perform similarly to a VAE+LDPC (LDPC is a classic error correcting code) setup.

First of all, the paper is quite well written and easily readable. Great work on explaining the motivation and the model -- the writing is clear and explains background knowledge extremely well.

The main contribution in the model is the use of discrete binary latents, instead of the standard continuous latents in a VAE. However, I am uncertain about the novelty of this contribution. There have been numerous works examining discrete latent variables in autoencoders (a random sampling: [1, 2, 3, 4]) and beyond. Furthermore, the method of training through discrete latents is also standard (VIMCO, though one can also imagine using more recent advances like REBAR or RELAX). The only difference would be the addition of noise to the discrete. I would be curious to see how that compares to recent works that have also added noise to discrete latents [5].

Thus, it strikes me that the main contribution of this work would be in comparing against the current best techniques for coding. However, the experiments section is weak, and does not provide significant evidence that the NECST model is better than the alternatives. NECST outperforms JPEG+ideal channel coding, but doesn't do much better than a VAE+LDPC baseline. This suggests that most of the gains comes from the encoder (source coding) model q(\hat{y} | x), instead of the joint training of source coding and error correcting code. It is not surprising that using a neural network to generate codes would provide significant gains. It's not clear that error correcting code aspect (noise in the latents) is particularly important.

Furthermore, in the classification results, the MLP model trained on the discrete codes gets 93% accuracy on noiseless MNIST inputs. You can easily get this accuracy by training logistic regression directly on the pixels. Despite what the authors write, this result suggests that the codes are not very useful for downstream learning. Furthermore, it is unclear why adding random noise to the inputs would significantly improve some of the weaker classifiers. The only reason I can think of is data augmentation, but this has nothing to do with the NECST model.

In conclusion, this is a well written paper, but the novelty is not apparent and the experimental results are weak, and so I am not convinced this is suitable for ICLR.

Additional Questions:
* How is the runtime computed? Specifically, for NECST, do you batch the data and then divide the forward pass time by the batch size? If this is how runtime is computed, it's not surprising that NECST does better, given that batching is cheap with modern hardware. If the actual forward pass time for a single example is cheaper than that of LDPC's belief propagation, then that would be quite promising.
* The authors state that VAEs optimize a lower bound on the marginal log-likelihood p(X), whereas NECST optimizes a lower bound on the mutual information I(X, Y), where Y is the noised code. The authors however do not discuss why one should optimize for mutual information compared to marginal log-likelihood. What are the advantages and disadvantages between the two?

[1] Semi-Supervised Learning with Deep Generative Models (https://arxiv.org/abs/1406.5298)
[2] Discrete Variational Autoencoders (https://arxiv.org/abs/1609.02200)
[3] Neural Discrete Representation Learning (https://arxiv.org/abs/1711.00937)
[4] Discrete Autoencoders for Sequence Models  (https://arxiv.org/abs/1801.09797)
[5] Theory and Experiments on Vector Quantized Autoencoders (https://arxiv.org/pdf/1805.11063.pdf)

---

> ### Author Response · Authors · 2018-11-08
> **clarification of the paper's main contribution and runtime experiment**
>
> We thank the reviewer for their insightful comments and feedback regarding the paper! We address the main points of the reviewer’s concern below:
>
> 1. Questionable novelty of discrete latent variable modeling: The novelty in this paper is our approach in addressing the joint-source channel coding (JSC) problem. The fact that NECST can be seen as a generative model with discrete latent variables is a novel observation from this paper. Precisely because we are **not in the traditional generative modeling setting**, our model : (1) includes latent additive noise (a discrete bit-flipping procedure), which to the best of our knowledge has not been done before, and (2) does not use a prior distribution as in a VAE-like setup [refs 1-3] while also not relying on additional discretization techniques [ref 4]. NECST is not comparable with [ref 5] because we want hard assignments of our data to the latent codes, as opposed to a mixture of codes. We also could not find any mention of adding noise to discrete latents in [ref 5] -- could the reviewer clarify this point?
>
> Although we build on prior work on discrete latent variable models, our (a) motivating application and (b) resulting training objectives are fundamentally different. Refs 1-5 do not address the problem of JSC. For example, it is unclear as to how these methods would address the problem of adding redundancy (when |z| is very large) to achieve robustness. For example, when we train a model on MNIST designed specifically for representation learning (discrete VAE) and evaluate for JSC (channel noise = 0.2), we find that the distortion increases by a factor of 3x.
>
> 2. Weak experimental section:
> 2a) Minor improvements against VAE + LDPC baseline: We would like to highlight that (1) we do see small performance improvements over VAE + LDPC in almost all cases, and (2) our method is orders of magnitude faster as we do not need to run multiple iterations of belief propagation for decoding. Also, LDPC is not a trivial baseline - it is an industry standard optimized over decades.
> 2b) Classification on noisy + noiseless MNIST: The purpose of the downstream classification experiment was to demonstrate that the latent codes, when trained with simulated channel noise, become more robust (“more useful”) for downstream tasks. This is in itself a novel observation: one would not normally think of JSC as a feature extractor. Specifically, when the latent codes are corrupted by the channel, the codes will be “better separated” in latent space so that the model will still be able to reconstruct accurately despite the added noise. Thus NECST can also be thought of as a “denoising autoencoder”-style method for learning more robust latent features with the added twist that the noise is injected into the latent space as opposed to the data space.
>
> 3. Runtime computation: The reviewer pointed out that batching the forward pass for the NECST decoder would lead to easy gains in speed. This is indeed the setup that we used, as we believe that NECST’s ability to allow for batching in the encoding/decoding process serves as an advantage of our model. But for a more fair comparison, we have re-run the timing experiment without batching the inputs (decoding one codeword at a time). We find that NECST still outperforms traditional LDPC decoding. Specifically, CPU is *slightly* faster than GPU (same order of magnitude), while NECST decoding still remains an order of magnitude faster than LDPC without batching.
>
> MNIST
> channel noise:	 0.0		0.1		0.2		0.3	 0.4		0.5
> LDPC:		 ['3.42E-04s', '1.11E-02s', '1.11E-02s', '1.11E-02s', '1.11E-02s', '8.11E-03s']
> NECST (CPU):	 ['4.42E-04s', '4.35E-04s', '4.32E-04s', '4.35E-04s', '4.45E-04s', '4.40E-04s']
>
> OMNIGLOT
> channel noise:	 0.0		0.1		0.2		0.3	 0.4		0.5
> LDPC:		 ['3.34E-04s', '1.15E-02s', '1.13E-02s', '1.17E-02s', '1.18E-02s', '7.95E-03s']
> NECST (CPU):	 ['4.44E-04s', '4.43E-04s', '4.44E-04s', '4.39E-04s', '4.44E-04s', '4.39E-04s']
>
> RANDOM
> channel noise:	 0.0		0.1		0.2		0.3	 0.4		0.5
> LDPC:		 ['6.80E-05s', '1.09E-03s', '1.24E-03s', '1.21E-03s', '1.16E-03s', '7.28E-04s']
> NECST (CPU):	 ['3.57E-04s', '3.87E-04s', '3.72E-04s', '3.63E-04s', '3.69E-04s', '3.67E-04s']
>
> 4. Choice of optimizing a lower bound on I(X,Y): We note that there seems to be a misunderstanding: as we are not in a generative modeling setup, we need a different objective from the standard ELBO as in a VAE. In JSC, we want to maximize the amount of error-free information that can be transmitted over our noisy channel. This is by definition (see MacKay: http://www.inference.org.uk/itprnn/book.html) the channel capacity, or the maximum mutual information I(X,Y) between our data X and noisy codes Y. Optimizing this lower bound on I(X,Y) (as computing the true objective is intractable) also has the nice and novel interpretation for NECST that allows us to view the framework from a generative modeling perspective.

---

### Official Review · AnonReviewer1 · 2018-11-11
**Interesting, yet limited paper**

**Rating:** 6
**Confidence:** 5

**Review:**

The authors set out to tackle an old problem (joint source-channel coding) with a principled approach and a fresh perspective. However, I find the paper quite limited both in terms of modeling choices as well as evaluation methodology. Specifically:

- The mutual information maximization approach is appropriate, but hardly novel. Besides being highly related to ELBO maximization, there have been several recent papers on rate-distortion optimization, as well as on deriving variational bounds for MI (see, for instance, Alemi et al.).

- The experimental setup is somewhat niche: in the context of image compression, both the fixed-rate constraint as well as the use of a binary symmetric channel are unusual. The vast majority of image compression methods are variable-rate, and for good reason: generic images tend to carry vastly different amounts of self-information, such that a fixed-rate code is almost guaranteed to achieve suboptimal *average* performance in terms of rate-distortion. Additionally, the vast majority of images today are sent over channels that already perform error correction, such as packet-switched networks (e.g., the Internet) or digital storage media, so that it's unclear why this particular case of joint source-channel coding would be practically relevant.

- I find the claim that the model is "competitive against industry standard compression" hardly justified based on the presented data. First, JPEG is now almost 40 years old. Since its inception, newer industry standards have exceeded it multiple times over in terms of rate-distortion performance. Second, JPEG was designed as a compression method for generic images. Comparing its performance on Omniglot and CelebA datasets is unfair, because the presented model can be trained to exploit special probabilistic structure in these datasets, while JPEG cannot. A widely used and accessible dataset better suited to compare against exisiting image compression methods would be the Kodak set, for example. And third, as explained above, JPEG is a variable-rate compression algorithm. How exactly were the number of bits required for JPEG to achieve the same distortion as NECST computed? To produce the plot in Figure 1, did the authors first compute an average rate for each average distortion, or was the computation done for each individual image, and then averaged to produce Figure 1 in a second step? This distinction could make a big difference.

- Regarding Sections 5.3 and 5.4: Could the authors please justify why they just double the length of the VAE representation? Wouldn't it be fairer towards LDPC to compare NECST to a VAE+LDPC code with various amounts of redundancy? Similarly, could the authors please justify comparing runtime only against a fixed 50 iterations of LDPC, rather than comparing against a range of possible values to make sure they are giving LDPC the benefit of the doubt?

---

> ### Author Response · Authors · 2018-11-18
> **justifications and modifications**
>
> We thank the reviewer for their insightful comments and feedback! We address the main points below:
>
> 1. Novelty of MI maximization: We agree that there has been a lot of work on (variational) MI maximization and R-D optimization, and we have added several relevant citations (Hinton & van Camp 1993, Honkela & Valpora 2004, Alemi et al. 2017, Chen et al. 2016, Ballé et al. 2016), into our paper (Section 4.1, paragraph 2). However, to the best of our knowledge none of these works address the problem of JSCC, and we view this paper as building on the existing body of work on image transmission.
>
> 2. Niche setup: We agree in that allowing the model to handle variable-rate image compression is important, and plan for this extension in future work. For this paper, we decided to first evaluate the feasibility of using fixed-rate codes and reported our findings as we think (1) our observations and (2) this framework’s connection with generative models (with are “fixed-rate”) could be valuable to the community.
> Regarding the setup’s relevance, we would like to note that even in wireless image/video transmission, JSCC is an important and active area of research in the systems/IEEE media communities. While there has been a lot of work in the early 2000’s [Cai et al. 2000, Mohr et al. 2000, Wu et al. 2005, Bi et al. 2014], JSCC was considered to be a very difficult problem. It was not until recently with the rise of deep learning that JSCC has garnered interest again in text/image modeling [Rao et al. 2018, Bourtsoulatze et al. 2018]. NECST serves as a way to model this entire communication process; specifically, our model’s implicit addition of redundancy into the latent codes is analogous to the forward error-correcting (FEC) techniques typically applied across/within packets in packet-switched networks [Zhai and Katsaggelos 2007]. Although more work and research is needed to turn these ideas into a deployable system, our work demonstrates the feasibility of an end-to-end neural network approach.
>
> 3. Experimental Details:
> 3a) Why JPEG: We found that JPEG performs quite well for small images, as most of the improvements in JPEG2000 and WebP come from larger block sizes (larger images). Additionally, as WebP only supports RGB images (https://developers.google.com/speed/webp/faq), we were unable to use it for grayscale images as it led to inferior compression. Thus we decided to use JPEG across all our image datasets. As a final note, we exclude the bytes that are shared among an image dataset when comparing to the number of bits used by NECST to make the comparison as fair as possible.
> 3b) Unfair JPEG vs. NECST compression: We would like to highlight that NECST’s ability to leverage statistical structure in images is an advantage when training data is available. Additionally, we discard a significant chunk of bytes of the compressed JPEG image that are shared across a dataset when comparing the image sizes, to try to make the comparison as fair as possible.
> With regards to the Kodak dataset, we are unable to validate our model on it because it would require a new dataset comprised of images with the same dimension, drawn from the same distribution as that of the Kodak images. This would also require a new architectural adjustment to NECST, as our framework is unable to handle the images of very high resolution. We plan to explore such improvements upon our model for future work. We have also modified our sentence in the Section 8 to make it more clear that our model is “competitive against a combination of industry standard compression and error-correcting codes.”
> 3c) Figure 1: We computed an average rate per individual image. As per Appendix D.1, we obtain the target distortion level by using a fixed bit-length budget with NECST, then use the JPEG compressor to encode the image at the desired level of distortion. We then use the resulting size of the image (ignoring headers, which we call f(d)) to obtain an estimate m = f(d)/C for the number of bits. After running this procedure over all images, we obtain an average at the very end.
>
> 4. Sections 5.3 / 5.4:
> 4a) VAE + LDPC comparison: We chose to double the length of the VAE representation such that we could work with a simple rate-1/2 LDPC code. In our initial experiments, where we fixed the number of LDPC bits (e.g. 200) and varied the number of bits used for the VAE, we found that there was not a significant difference in the results obtained.
> 4b) Runtime experiment: The BP implementation we use from Radford Neal is such that once the tentative decoding (based on bit-by-bit probabilities) is a valid codeword, the algorithm halts. Therefore, we implicitly allow the LDPC decoder to terminate as early as it wants. In the original LDPC package documentation, there were several examples provided where the LDPC decoder was run for a maximum of 200-250 iterations of BP. In initial experiments, we found that for our setup roughly 50 iterations as the max were sufficient.

---

> > ### Comment · AnonReviewer1 · 2018-11-29
> > **Issues remain**
> >
> > Thank you for providing these justifications, which I find mostly acceptable.
> >
> > Unfortunately, I find myself still dissatisfied with the experimental validation.
> >
> > The fact that Kodak can't be used due to larger image resolution is irrelevant here - I brought up Kodak as an example. The criticism was that Omniglot and CelebA are not generic images. If resolution is a problem, you could use CIFAR, for example, downsample Kodak, or use any other dataset that is not restricted to a particular class of images (such as faces or characters).
> >
> > I do not think the paper should be published with the statement that it is "competitive with industry standard compression". The provided data does not support this conclusion, because the comparison is not fair.
> >
> > To be clear, my current rating is contingent on either removing this statement in the camera-ready version, or providing better evaluation data.

---

> > > ### Author Response · Authors · 2018-11-29
> > > **proposed modification of statement**
> > >
> > > Thank you for the feedback. We agree that we have demonstrated our results on MNIST, CelebA, Omniglot, and SVHN, which are datasets where generative models such as variational autoencoders have been shown to work well. We are happy to revise the sentence:
> > >
> > > “We showed that the model: (1) is competitive against a combination of industry standard compression and error-correcting codes...”
> > > to
> > > “We showed that the model: (1) is competitive against a combination of JPEG and LDPC codes on Omniglot and CelebA...”
> > >
> > > in the final version of the paper. Alternatively, we are open to suggestions for a more appropriate wording.

---

> > > > ### Comment · AnonReviewer1 · 2018-12-10
> > > > **Agreed**
> > > >
> > > > Sorry for the delay –
> > > >
> > > > Thank you, qualifying the statement with the datasets like you suggested is acceptable.

---

### Meta-Review · Area_Chair1 · 2018-12-15

**Confidence:** 3
**Recommendation:** Reject

**Metareview:**

This paper proposes a principled solution to the problem of joint source-channel coding. The reviewers find the perspectives put forward in the paper refreshing and that the paper is well written. The background and motivation is explained really well.

However, reviewers found the paper limited in terms of modeling choices and evaluation methodology. One major flaw is that the experiments are limited to unrealistic datasets, and does not evaluate the method on a realistic benchmarks. It is also questioned whether the error-correcting aspect is practically relevant.